# Glycine Metabolism and Its Alterations in Obesity and Metabolic Diseases

**DOI:** 10.3390/nu11061356

**Published:** 2019-06-16

**Authors:** Anaïs Alves, Arthur Bassot, Anne-Laure Bulteau, Luciano Pirola, Béatrice Morio

**Affiliations:** 1Université de Lyon, Laboratoire de Recherche en Cardiovasculaire Métabolisme Diabétologie et Nutrition, Institut National de la Santé et de la Recherche Médicale, Institut National de la Recherche Agronomique, Institut National des Sciences Appliquées de Lyon, Université Claude Bernard Lyon 1, Unite Mixte de Recherche 1060, 69310 Pierre Bénite, France; anais.alves@etu.univ-lyon1.fr (A.A.); arthur.bassot@etu.univ-lyon1.fr (A.B.); luciano.pirola@inserm.fr (L.P.); 2Université de Lyon, Institut de Génomique Fonctionnelle de Lyon, Ecole Normale Superieure de Lyon, Centre National de la Recherche Scientifique, Université Claude Bernard Lyon 1, Unite Mixte de Recherche 5242, 69007 Lyon, France; anne-laure.bulteau@ens-lyon.fr; 3INRA U1397-INSERM U1060—Laboratoire CarMeN, Hôpital Lyon Sud secteur 2, bâtiment Cens-Eli D, 165 Chemin du Grand Revoyet, 69310 Pierre Bénite, France

**Keywords:** amino acid metabolism, gut–liver axis, pathophysiology of metabolic disorders, nutritional prevention

## Abstract

Glycine is the proteinogenic amino-acid of lowest molecular weight, harboring a hydrogen atom as a side-chain. In addition to being a building-block for proteins, glycine is also required for multiple metabolic pathways, such as glutathione synthesis and regulation of one-carbon metabolism. Although generally viewed as a non-essential amino-acid, because it can be endogenously synthesized to a certain extent, glycine has also been suggested as a conditionally essential amino acid. In metabolic disorders associated with obesity, type 2 diabetes (T2DM), and non-alcoholic fatty liver disease (NAFLDs), lower circulating glycine levels have been consistently observed, and clinical studies suggest the existence of beneficial effects induced by glycine supplementation. The present review aims at synthesizing the recent advances in glycine metabolism, pinpointing its main metabolic pathways, identifying the causes leading to glycine deficiency—especially in obesity and associated metabolic disorders—and evaluating the potential benefits of increasing glycine availability to curb the progression of obesity and obesity-related metabolic disturbances. This study focuses on the importance of diet, gut microbiota, and liver metabolism in determining glycine availability in obesity and associated metabolic disorders.

## 1. Introduction

Glycine is the protein-forming amino acid with the smallest molecular weight (NH_2_-CH_2_-COOH; 75.067 g/mol). The hydrogen in the glycine’s side chain allows it to integrate both hydrophilic and hydrophobic environments within the polypeptide chain. Although glycine can be endogenously produced, studies in several animal models support the notion that glycine may be conditionally essential [1], as the amount of glycine synthesized in vivo may sometimes be insufficient to meet the organism’s metabolic needs [2,3,4]. Chronic glycine insufficiency may impact health status in the long term. Indeed, glycine, besides being a protein’s building block, is also a bioactive amino acid that participates in the regulation of gene expression [5,6], protein configuration and activity [6], and several biological functions, such as glutathione synthesis [7]. Low plasma glycine concentrations have been consistently reported in association with obesity, T2DM, and NAFLDs [8,9]. These observations suggest that in the long term, mild deficiency in glycine may participate in the etiology of metabolic diseases (Table 1). Therefore, the aim of this review is to summarize the evidence supporting this hypothesis.

The following points will be emphasized: (1) situating glycine metabolism in a context of the physiopathology of metabolic diseases associated with obesity (insulin resistance, T2DM, and NAFLDs); (2) understanding the potential causes of decreased glycine availability in these situations; and (3) summarizing the current knowledge on the benefits of improving glycine availability through dietary supplementation. The present review broadens the scope of previously published reviews [10,11,12,13] by integrating the importance of dietary patterns, the contribution of the gut microbiota, and the interaction with host metabolism in determining glycine availability.

## 2. Glycine Dietary Intake and Metabolism

Glycine dietary intake varies from 1.5 to 3 g/day depending on the protein intake of the individual. Glycine content in the protein fraction of different dietary sources is relatively uniform, except for rice, which is about twice as rich in glycine as compared to other proteins from animal or vegetable origins [21]. The multicentric European Prospective Investigation into Cancer and Nutrition (EPIC) study has shown that the average daily intake of glycine (adjusted for age, body mass index, smoking status, and alcohol intake) varies from 2.28 to 3.12 g/day in adult males with different patterns of dietary protein sources (meat or fish eaters, vegetarians or vegans) [22]. The interaction between glycine dietary intake and its endogenous biosynthetic and catabolic pathways is crucial in determining the ability of an individual to meet their needs (Figure 1). To begin with, we will first briefly examine the biochemical principles and highlight the relative importance of the main endogenous pathways of glycine synthesis and degradation. These pathways have been recently thoroughly reviewed by Adeva-Andany et al. [11].

### 2.1. Glycine Synthesis

Using isotopic tracers in young men, whole body glycine flux was estimated to average 34–35 mg/kg/h in the fed state [2,23]. Of this systemic flux, 35% of the glycine comes from endogenous synthesis [3]. In the post-absorptive state, whole body glycine flux is decreased by half and reaches around 18 mg/kg/h [23,24], with de novo synthesized glycine contributing to 81% of the systemic flux [24]. The average rate of whole body glycine de novo synthesis could, thus, be estimated at 12 to 15 mg/kg/h. This rate can be decreased when non-essential amino acid intake is reduced [3], but it is only slightly affected by total amino acid intake [3], hyperglycemia, or hyperinsulinemia [24,25].

Glycine is mainly synthesized from serine, threonine (in most mammals but not in humans; [26]), choline, sarcosine (*N*-methylglycine), and glyoxylate, and during endogenous synthesis of L-carnitine (Figure 1). Theoretical calculations have determined that serine and its precursors are the main contributors to the endogenous production of glycine. Serine contributes to the synthesis of approximately 2.5 g of glycine/d, which is close to the mean dietary intake of glycine discussed above [4]. Collectively, the other biosynthetic pathways contribute to the endogenous production of glycine to a much lesser extent (less than 15% of the serine contribution) [4].

In healthy individuals, glycine is inter-convertible with serine (Figure 1). Serine is mostly derived from the diet, but it can also be produced from glucose via 3-phosphoglycerate, especially in kidneys. Glycine synthesis from serine is compartmentalized, being catalyzed by serine hydroxymethyltransferase 1 (SHMT1) in the cytosol and by SHMT2 in the mitochondrial matrix [27]. While SHMTs are ubiquitously expressed, most of the SHMT-dependent glycine synthesis occurs within the liver, mainly via mitochondrial SHMT2. A prior study indeed showed that mitochondrial isoform of SHMT and intracellular serine concentration regulate the rate of glycine synthesis in mutant lines of cultured Chinese hamster ovary cells [28]. SHMT activity requires two cofactors, pyridoxal phosphate and tetrahydrofolate (THF). Theoretical calculations suggest that the stoichiometry of this reaction may be the limiting step of glycine synthesis and may explain why the in vivo synthesis rate of glycine may become insufficient to meet the metabolic needs [29]. More specifically, stoichiometry of the reaction catalyzed by SHMT requires glycine and 5,10-methylene tetrahydrofolate (CH_2_-THF) to be produced in equimolar amounts, regardless of differences in the metabolic needs for the two molecules. Since the methylene unit (CH_2_) must be released from CH_2_-THF before THF can be reused for glycine synthesis [30], the flux of methyl transfers must match the needs for glycine synthesis and also that of glycine catabolism by the cleavage system. SHMT in human placenta has been pointed out as an important enzyme to cover glycine needs for fetal growth [31]. However, alterations in the rate of conversion of serine to glycine have not been reported yet in obesity and associated metabolic disorders. Decrease in the plasma level of serine reported in those conditions [30,32], although not consistently [9], may suggest a potential reduction in the activity of this pathway.

Glycine synthesis from the choline biosynthetic pathway [33] is not predominant as dietary choline intake is very low (400–500 mg/d). However, this pathway is relevant because it also intervenes in the regulation of methyl donor availability, and consequently, in cellular methylation processes. Choline-dependent glycine biosynthesis involves the intermediary metabolites betaine (trimethylglycine), dimethylglycine, and sarcosine (*N*-methylglycine), as illustrated in Figure 1. The enzymes regulating the methyl transfers in the production of glycine from betaine are successively the cytosolic enzyme betaine-homocysteine S-methyltransferase (BHMT), and the mitochondrial enzymes dimethylglycine dehydrogenase (DMGDH) and sarcosine dehydrogenase (SDH). Interestingly, expression of BHMT, which catalyzes the methyl transfer from betaine to homocysteine to produce dimethylglycine and methionine, was shown to be upregulated in the liver of high-fat fed mice [34,35].

In the liver, glycine can also be produced from the conversion of glyoxylate by alanine:glyoxylate aminotransferase (AGXT), which simultaneously converts alanine to pyruvate [36]. AGXT is mainly present in peroxisomes in humans and plays a key role in limiting the rate of oxalate synthesis. Glyoxylate is produced as a byproduct of the pentose phosphate pathway or the breakdown of serine and hydroxyproline.

The enzyme dihydrofolate reductase (DHFR) is a key enzyme in cell replication. It catalyzes the recovery of tetrahydrofolate, which is an essential reaction for the de novo synthesis of glycine, among others [37]. Interestingly, isolation of Chinese hamster cell mutants showed that mutant cells that no longer express DHFR require glycine to survive [38]. Downregulation of DHFR protein expression and of metabolites from the vitamin B group have been reported in the liver of high-fat fed mice [39]. In addition, vascular DHFR protein content was decreased in db/db mice compared with db/m mice [40].

Finally, hepatic de novo synthesis of glycine is catalyzed by glycine synthase, also called glycine cleavage enzyme [41]. This enzymatic system is composed of four mitochondrial protein components: P protein (a pyridoxal phosphate-dependent glycine decarboxylase), H protein (a lipoic acid-containing protein), T protein (a tetrahydrofolate-requiring enzyme), and L protein (a lipoamide dehydrogenase). Glycine synthase catalysis is reversible and participates in the hepatic catabolism of glycine (see below) (Figure 1).

### 2.2. Glycine Catabolism

Dietary glycine is rapidly converted to serine by the cytosolic SHMT1. This pathway accounts for almost half of the whole body glycine flux [23] and is a quantitatively major acceptor of methyl groups from CH_2_-THF. In young men, tracking the fate of orally administered ^15^N-glycine showed that the glycine nitrogen group is mainly transferred to serine (54%), but also to urea (20%), glutamine/glutamate (15%), alanine (7%), and other amino acids (leucine, isoleucine, valine, ornithine, proline, and methionine) for the remaining extent [42]. This shows that the nitrogen group from glycine or serine participates in multiple transamination reactions via glutamate formation. Using ^13^C-glycine infusion, it was observed that the flux through SHMT to ^13^C-serine was decreased in the liver of obese rats with nonalcoholic steatohepatitis (NASH) [43], thus suggesting that hepatic SHMT activity may be altered in obesity and related-metabolic disorders. To support this observation, gene expression of SHMT1 and 2 was found to be downregulated in the liver of patients with NAFLD [30].

The second main pathway of glycine utilization involves the production of CO_2_ and NH_4_^+^ by the reverse reaction of the glycine synthase or glycine cleavage system mentioned above [41,44] (Figure 1). This enzymatic conversion is physiologically relevant as it provides CH_2_-THF. CH_2_-THF is the major methyl group donor, through S-adenosylmethionine (SAM), for the biosynthesis of molecules such as purines, thymidylate, and methionine [41]. Exploration of the rate of glycine oxidation and urea synthesis in patients with NASH revealed no significant alterations compared to healthy individuals in the fasted state [45].

Other minor pathways include the conversion of glycine to glyoxylate by D-amino acid oxidase, to glycocyamine by the arginine:glycine amidinotransferase (AGAT), and to sarcosine by the glycine N-methyltransferase (GNMT) [41,44]. AGAT is present in the cytosol and in the mitochondrial intermembrane space and synthesizes the precursor of creatine [46], mainly in the kidney, liver, and pancreas [47]. GNMT is localized mainly in the cytosol of periportal hepatocytes and in the exocrine pancreas. The glycine-to-sarcosine conversion requires SAM, and thus plays a key role in regulating methylation processes [5,6]. Importantly, GNMT expression is decreased both in the liver of animal models receiving a high-fat diet [48] and in patients with hepatic steatosis [49]. These alterations are further developed in Section 6.3.

### 2.3. Glycine Uptake

Together with de novo glycine synthesis, uptake of dietary glycine and re-absorption from the kidney represent the other main route for glycine bioavailability. Three classes of glycine transporters have been described: SLC36 gene family (PATs) in the intestine; SLC6 (GLYTs) in the intestine, kidney, and neural tissues; and the SLC38 family, with a wide tissue distribution [50]. The proton/amino acid transporters PAT1 and PAT2, expressed on the apical membrane of intestinal epithelial cells, mediate the symport of protons and small neutral amino acids, including glycine [50].

The sodium/chloride-dependent transporters GlyT1 and GlyT2 were first cloned from the rat brain [51,52], where glycine is an important modulator of neurotransmission [53]. GLYT1 is responsible for the high affinity transport of glycine and its derivatives, and is inhibited by sarcosine [54]. Selective GlyT-1 inhibitors increase extracellular glycine levels and potentiate N-methyl-D-aspartate (NMDA) receptor activity [55]. Hence, GLYT1 inhibition in the dorsal vagal complex suppressed hepatic glucose production, increased glucose tolerance, and reduced food intake and body weight gain in healthy, obese, and diabetic rats [56]. GLYT1 is also found throughout the intestine, where it is responsible for 30–50% of glycine uptake into intestinal epithelial cells across the basolateral membrane. It recently appeared to function to maintain glycine supply to enterocytes and colonocytes, mediating cytoprotection in the intestinal absorptive cells [57]. GlyT-2 has a lower affinity for glycine compared to GLYT1 [58]. It is involved in particular in maintaining terminal supplies of glycine for inhibitory glycinergic neurotransmission [59].

Na^+^-dependent transporters of the SLC38 family are ubiquitously expressed, and are particularly enriched in dividing cells and in cells types actively engaged in amino acid metabolism, such as hepatocytes, kidney cells, and neurons. Expression of the SLC38 transporters is polarized, being limited to plasma membrane regions facing blood vessels or engaged in cell–cell contacts, and not occurring in the apical membrane of absorptive epithelia [60].

### 2.4. Glycine Conjugation and Excretion

Recent evidence suggests that the glycine conjugation pathway is an essential detoxification pathway [61,62,63]. Glycine can be conjugated to various endogenous and xenobiotic metabolites (e.g., benzoate, derivatives of branched chain amino acids (BCAA), β-oxidation intermediates and metabolites of polyphenols), which can be potentially toxic when they accumulate in the organism [62]. The resulting acylglycines are less toxic and more hydrophilic and are excreted in the urine [62,64]. As these metabolites are esterified to CoA, it has been proposed that glycine conjugation contributes to CoA homeostasis, since the reaction releases CoA [64]. Conjugation activity depends on the metabolite and on the enzyme catalyzing the reaction, i.e., acid:CoA ligases and glycine N-acyltransferases (GLYAT) [64,65].

Glycine is also involved in the enterohepatic cycle of bile acids (BAs), which are required for lipid absorption and regulation of cholesterol homeostasis [66]. These water-soluble steroids are synthesized from cholesterol exclusively in hepatocytes and include cholic acid and chenodeoxycholic acid [66]. Physiologically, most BAs are conjugated to either glycine or taurine, forming glycocholic acid and taurocholic acid, respectively [66]. In humans, glycine conjugation dominates in adulthood [67]. It is about 3.5 times higher than taurine and depends on the dietary amino acid intake [68]. Bile acid-coenzyme A: amino acid N-acyltransferase (BAAT) is the only enzyme involved in the 2 types of conjugation [67,69]. This enzyme resides in peroxisomes of hepatocytes [67,69]. After conjugation with glycine or taurine, BAs are secreted in bile and stored in the gall bladder. After food intake, BAs participates in digestion and fat absorption thanks to their detergent properties. Thereafter, BAs and associated amino acids are reabsorbed in the intestine, while the associated molecules are secreted in the feces. Obesity has been associated with increased hepatic bile acid synthesis [70] and fifty years ago, active transport of conjugated bile salts was shown to be enhanced in streptozotocin-diabetic rat small intestine [71].

## 3. Plasma Concentrations of Glycine are Decreased in Obesity and Associated Metabolic Disorders, Although Dietary Intake are Unaltered

Physiological glycine plasma concentration ranges between 200 and 300 µmol/L. Plasma glycine levels were reported to be 9 to 13% lower in patients with NAFLD, whether obese or not, compared to controls [9]. Using the GSG (glutamate-serine-glycine) index, calculated as the ratio between the plasma level of glutamate and that of serine + glycine, this study found that the GSG index was correlated with an increased level of liver enzymes, particularly gamma-glutamyltransferase, and liver fibrosis. The study also suggested that decreased plasma glycine concentration was associated with hepatic insulin resistance. This finding is supported by a recent meta-analysis, which showed that plasma glycine concentration is consistently lower in patients with obesity and T2DM compared to healthy individuals (−11 and −15%, respectively) [72]. Another meta-analysis showed that plasma glycine concentration has a very significant inverse association with the risk of developing T2DM [8]. In addition to these observational studies, the contribution of circulating glycine to metabolic control has been demonstrated experimentally. Using the insulin resistance homeostasis model, glycine concentrations were found to be positively correlated with insulin sensitivity and inversely correlated with insulin resistance [19]. Importantly, among all plasmatic amino acids analyzed individually, glycine had the highest association with alteration in insulin sensitivity, measured using a hyperinsulinemic-euglycemic clamp in 399 non-diabetic subjects presenting a broad range of insulin sensitivity [15]. The same positive association was also found in a smaller study, which used a similar approach [20]. In a consistent fashion, plasma glycine concentration was found to be lower in the lean offspring of T2DM parents compared to healthy children of control subjects [73]. On the contrary, the improvement in insulin sensitivity observed after weight loss, exercise, or medical administration of metformin was associated with improved plasma glycine concentrations in obese or T2DM subjects [74,75,76,77,78]. Importantly, prospective studies indicate that a decrease in plasma glycine concentration is a strong predictive factor for incident glucose intolerance and T2DM [79,80,81].

Similar observations have been made with two glycine precursors, i.e., betaine (trimetylglycine) [82,83] and dimethylglycine [84]. Of note, plasma betaine levels are variable, ranging from 20 to 60 µM in women and from 25 to 75 µM in men, whereas plasma dimethylglycine levels are typically below 10 µM [85]. Plasma betaine levels were lower in obese men [86] and in insulin-resistant individuals [87] as compared to healthy people. Hence, plasma betaine levels were positively correlated with insulin sensitivity [87]. In the multi-ethnic clinical trial Diabetes Prevention Program, 3234 obese or overweight individuals at high risk of developing T2DM were followed over an average of 3.2 years. In this cohort, baseline plasma betaine levels were significantly associated with incident T2DM, and any increase of betaine following a 2-year management (either through lifestyle or metformin) was associated with a lower incidence of overt T2DM [88]. Finally, in 3621 non-diabetic persons followed for an average of 7.5 years, plasma betaine was inversely correlated in a significant manner with the onset of T2DM [89].

Somehow puzzling, however, was that a reverse correlation was demonstrated with respect to sarcosine, which is an intermediate in the biosynthesis and degradation of glycine. Indeed, sarcosine plasma concentrations were increased in patients with metabolic syndrome as compared to healthy controls, and inversely correlated with the decrease in plasma glycine [90]. This may be due to alterations in metabolite urinary excretion. Indeed, in agreement with Schartum-Hansen et al. [91], Svingen et al. [89] found positive associations between the incidence of T2DM and increased urinary excretion of metabolites related to glycine metabolism, including betaine, dimethylglycine, and sarcosine.

## 4. Potential Causes of Decreased Glycine Availability

Here, we put forward the hypothesis that a decrease in glycine plasma concentrations in obesity, and secondarily in insulin resistance and T2DM, is due to a reduction of the amino acid availability resulting from the simultaneous contribution of three distinct mechanisms: (i) decreased gut absorption; (ii) decreased biosynthesis; and (iii) increased catabolism or urine excretion. We review here the potential mechanisms contributing to the establishment of glycine deficiency during the progression of metabolic disorders associated with obesity (Figure 2).

### 4.1. Importance of Dietary Patterns in Determining Glycine Availability

Based on the hypothesis that the amino acid intake determines the amino acid bioavailability and health impact, the prospective Tehran Lipid and Glucose Study examined the relationship between amino acid consumption and risk of pre-diabetes in 1878 people without T2DM, followed for an average of 5.8 years [92]. Amino acid consumption was stratified into three groups: (1) high intake of lysine, methionine, valine, aspartic acids, tyrosine, threonine, isoleucine, leucine, alanine, histidine, and serine; (2) high intake of glycine, cysteine, arginine, and tryptophan; and (3) high intake of proline and glutamic acid. Total protein consumption and high consumption of amino acids from group 1 were not associated with the development of pre-diabetes. Furthermore, high consumption of amino acids from group 2 only tended to be negatively associated with incident pre-diabetes, whereas a high consumption of amino acids from group 3 was associated with an increased risk of prediabetes [82]. This suggests that dietary intake of amino acids may selectively contribute to the risk of prediabetes (or protection thereof), depending on the amino acids being ingested.

To support this hypothesis, the multicentric EPIC study has shown that, among adult European males, the average daily intake of glycine varies little according to food choices (meat or fish eaters, vegetarians or vegans) [22]. By contrast, plasma glycine concentration was significantly altered according to food choice, being higher in the vegetarian and vegan groups compared to the meat eaters. Therefore, one can suppose that consumption of meat proteins may be associated with a low level of circulating glycine, whereas consumption of proteins from vegetable sources may be associated with higher glycine concentrations. This hypothesis, while requiring further clinical and experimental validation, is in agreement with findings from a randomized clinical trial showing that glycine concentrations after one week of a meat diet were slightly lower than those observed after a vegetable protein diet, even though the amount of glycine provided by the meat diet was approximately 50% higher as compared to that provided by the vegetarian diet [93]. The fact that low circulating glycine is associated to metabolic diseasesis also consistent with findings from an observational study [94] and prospective studies [95]. An investigation conducted on 2681 individuals of the sub-cohort European Prospective Investigation into Cancer and Nutrition (EPIC)-Potsdam [95] showed that total consumption of red meat was significantly associated with the risk of T2DM, after adjustment for age, sex, lifestyle, diet, and body mass index. Six plasma biomarkers, including decreased glycine, were associated with red meat consumption and risk of T2DM. Higher circulating glycine was associated with reduced risk of T2DM and was negatively correlated with red meat consumption. These results cannot, however, prove the causality of the associations. Indeed, results were not adjusted for dietary intake of glycine precursors, such as betaine, which is highly enriched in seeds, cereals, beets, and spinach. In the Complex Diseases in the Newfoundland Population: Environment and Genetics (CODING) study, betaine dietary intake in 2394 adults was the lowest in the group with the highest index of insulin resistance, and the highest in the group with the lowest index of insulin resistance [96]. Furthermore, a randomized clinical trial from Beaumont et al. [97] failed to show alterations in plasma glycine concentration in response to a 3-week dietary supplementation with casein versus soy protein in overweight humans.

To conclude, food consumption, in terms of dietary patterns, rather than dietary glycine intake, is a key determinant of glycine availability, potentially through modulation of glycine endogenous metabolic pathways.

### 4.2. Contribution of the Gut Microbiota in Determining Glycine Bioavailability

It is well-known that the liver is the main site of glycine degradation, so one could speculate that alteration in that component could be the main cause of alteration in glycine availability. However, a significant amount of glycine is also catabolized in the small intestine [10]. In young pigs, about 30% of dietary glycine is used or degraded in the small intestine by the microbiota, and is therefore not available systemically [98,99,100]. Indeed, glycine is highly incorporated into proteins of both gram-positive and gram-negative gut bacteria [100]. This indicates that glycine is an important amino acid for supporting the optimal growth of the gut microbiota. In patients with T2DM not treated with metformin, Forslund et al. [101] have shown that the gut microbiota was enriched in genes involved in glycine degradation. Thus, imbalances in the use of amino acids by the gut microbiota—particularly glycine—could affect the biological function of the host. However, the contribution of bacterial species from different taxa to the overall determination of glycine metabolism has not yet been studied in detail, especially in humans.

Therefore, a relevant and still unanswered question relates to the potential involvement of the gut microbiota in enhancing glycine availability for the host [102]. An interesting study showed that in rats fed with β-glucans, probiotic supplementation with *Lactobacillus plantarum HEAL 9* and *19* increased portal and peripheral glycine concentrations [103]. In humans, dietary supplementation with *Lactobacillus acidophilus*, *Bifidobacterium longum*, and fructo-oligosaccharides reduced the fecal excretion of glycine [104]. Reciprocally, administration of species curbing the bioavailability of glycine has also been reported [100]; for example, introduction of *Lactobacillus paracasei* in the gut of germ-free mice reduced glycine concentrations in the jejunum and ileum [105]. This suggests that utilization or biosynthesis of glycine can differ between bacterial species within the gut and that pre- and probiotic dietary interventions can be used to manipulate glycine availability in the intestine, and secondarily, systemic availability. At present, however, there are too few studies available to provide unequivocal and strong support to the notion that administration of selected bacterial strains may be capable of enhancing glycine availability for the host.

Another putative action of the microbiota may occur through its capability to affect the host glycine metabolism [106]. A recent study in rats showed that the quality of the food matrix (liquid fine or gelled coarse emulsions) with similar amino acid composition alters the microbial composition in the ileum and caecum, and induces changes in the expression of amino acid transporters, including glycine transporters, in the ileum [107]. Therefore, the interactions between dietary proteins (and their presentation in the food matrix) and the gut microbiota can modulate the host’s glycine utilization.

Finally, bacterial glycine metabolism can vary along the intestinal tract according to changes in microbiota composition and richness [108]. In the colon, amino acids are no longer absorbed, but rather are intensively metabolized by the microbiota. In that context, especially under conditions of high pH and low carbohydrate availability, glycine is one of the amino acids readily catabolized by colonic bacteria, thereby contributing to the synthesis of various metabolites, such as ammonia, short-chain and branched-chain fatty acids, organic acids, gaseous compounds, and amines [100]. Those microbial metabolites can impact the microbiota metabolism, the physiology of the intestinal epithelium, and the host metabolism and health. Microbial metabolites can modulate gene expression in bacteria, thus altering the expression of enzymes involved in amino acid metabolism. In addition, an elegant study using germ-free versus conventionally-raised mice demonstrated that approximately 10% of the host transcriptome is regulated by microbial agents. The concerned pathways mainly comprise genes for immunity, cell proliferation, and metabolism, and their alteration is highly localized in gut compartments [109]. Interestingly, Mardinoglu et al. [110] showed that glycine bioavailability was lower in the gut of conventionally-raised mice compared to germ-free mice and that this decrease differentially modulated the intestinal expression of genes involved in glycine synthesis and metabolism, thus reducing its bioavailability in the portal vein. This suggests that the presence of the gut microbiota reduces the bioavailability of glycine for the liver.

Overall, these data suggest that gut microbiota dysbiosis likely plays a significant role in decreasing glycine availability in obesity and associated metabolic disorders.

### 4.3. Interaction between Host Metabolism and Fate of Glycine

*Hormonal regulation of glycine degradation:* Glycine plasma concentration is tightly regulated by glucagon, which is a major regulator of hepatic glycine metabolism. Glucagon stimulates the activity of the glycine cleavage enzyme and the degradation of glycine [111]. Hence, glucagon deficiency is associated with a rise in circulating glycine, whereas an excess of glucagon decreases it [112]. As obesity and associated metabolic disorders are characterized by increased plasma glucagon concentration [113], this pathway could participate in increased glycine degradation [63].

*Interaction with BCAA metabolism:* Impaired hepatic BCAA metabolism in obesity was recently shown to significantly contribute to the decrease in glycine circulating concentration [114]. In that context, a diet restricted in BCAA restored acylglycine level in urine and increased circulating glycine level given to a rodent model of obesity [114]. White et al. [114], in Zucker fatty versus lean rats, showed that the increased circulating concentration in BCAA that results from the reduced activity of the complex branched-chain keto acid dehydrogenase (BCKDH) in the liver [115,116] directly affects skeletal muscle amino acid metabolism following compensatory upregulation of BCKDH activity in that tissue. The authors elegantly showed that in obese animals, restricting BCAA dietary intake to reduce plasma BCAA levels partially to fully restored plasma and muscle glycine concentration, thus revealing a close relationship between glycine and BCAA metabolism. Yet, the underlying cellular mechanisms still remain to be fully elucidated.

*Glycine urinary excretion:* Decreased levels of circulating glycine could also reflect an increase in the urinary excretion of the glycine precursors betaine, dimethylglycine, and sarcosine, as observed in patients at high risk for T2DM [89,117]. This might suggest impaired tubular handling of glycine precursors. Indeed, patients at high risk for T2DM may present renal tubular dysfunction, even without overt renal impairment [118].

*Host genetics and glycine metabolism:* A mendelian randomization study showed a significant association between circulating levels of glycine and the single nucleotide polymorphism (SNP) rs715 in the 3′ untranslated region of the Carbamoyl-Phosphate Synthase 1 (*CPS1*) gene [119]. CPS1 is the rate-limiting enzyme for the urea cycle. It produces carbamoyl phosphate from ammonia (NH_4_^+^) and bicarbonate (HCO_3_^−^) to feed the urea cycle in the mitochondria, perhaps competing with the glycine cleavage system in synthesizing glycine. Association between the SNPs rs10206976 and rs12613336 in the *CPS1* gene were also pinpointed in a genome-wide association study (GWAS) [120], which suggested that individuals with a genotype leading to higher expression of *CPS1* have lower levels of circulating glycine, and the opposite also holds true. Recently, meta-analysis of GWAS in 30,118 individuals of European ancestry showed four significant loci associated with circulating glycine levels [121]: two loci at the glycine cleavage system (GCS) subunits, protein P (SNP rs71503800 at *GLDC*, glycine decarboxylase) and protein H (SNP rs11860711 at *GCSH*); and two novel loci at enzymes involved in serine metabolism SNP rs478093 in *PHGDH* (phosphoglycerate dehydrogenase) and SNP rs6955423 in *PSPH* (phosphoserine phosphatase). In addition, GWAS revealed an association between the levels of betaine and the SNPs s499368 in the *SLC6A12* gene (which codes for a betaine transporter) and rs17823642 near the *BHMT* gene [119], and between the levels of dimethylglycine and the SNP rs2431332 at the *DMGDH* locus (coding for dimethylglycine dehydrogenase) [84]. On the contrary, a genetic variant in *DMGDH*, causing lower DMGDH, was significantly associated with increased plasma insulin, increased HOMA index of insulin resistance, and increased risk of incident T2DM [84]. Given the paucity of available data, further studies are required to fully decipher the key steps linking alteration in the intermediate metabolism of glycine and the risks for developing metabolic disorders, such as T2DM.

## 5. Potential Benefits of Glycine Supplementation in Obesity and Associated Metabolic Disorders

Interventions that reverse or delay the onset of T2DM, such as bariatric surgery, are associated with increased glycine plasma concentrations [74,75,76,77,78]. In addition, acute glycine supplementation (5 g/day) was reported to improve insulin response and glucose tolerance [122,123]. Table 2 summarizes the findings of preclinical and clinical studies described in this paragraph. On a longer administration schedule (0.1 g of glycine/kg/day for 14 days in association with N-acetylcysteine), an improvement of insulin sensitivity in elderly patients with HIV was observed [124]. The health benefits of glycine supplementation were attributed to an improvement in glutathione synthesis and antioxidant protection. As a cautionary note, however, current conclusions cannot extend beyond a dose of 15 g of glycine per day, which is the highest dose well tolerated in adult humans [125].

Regarding the glycine precursor betaine, preclinical research showed that betaine supplementation improves insulin resistance and glucose homeostasis [87,126] and reduces liver fat deposition [34,87] in mice fed a high-fat diet. By contrast, a recent randomized controlled trial did not reveal major effects of betaine supplementation (3.30 g orally twice daily for 10 days, followed by 4.95 g twice daily for 12 weeks vs. placebo) on glucose homeostasis and liver fat deposition in patients with obesity and pre-diabetes [127]. Because betaine supplementation was associated with enhanced circulating levels of dimethylglycine, the authors suggested that betaine metabolism may be hampered by the activity of the enzyme DMGDH, which connects betaine to glycine synthesis by demethylating dimethylglycine into sarcosine. As discussed previously, allele variants at the *DMGDH* locus causing lower DMGDH enzyme levels were associated with insulin resistance [84].

Taken together, the above studies suggest that administration of glycine to appropriately selected patients with obesity or pre-diabetes may provide a nutritional approach to slow down the progression towards overt T2DM. However, long-term interventional studies are required to further explore the benefits of glycine supplementation in T2DM. Furthermore, the observation that glycine supplementation improves hepatic steatosis in animal models of obesity [128,129] should provide a motivation for human clinical testing.

## 6. The Contribution of Glycine to Host Metabolism and the Pathogenesis of Metabolic Disorders

### 6.1. Importance of Glycine for Antioxidant Protection by Glutathione

Glycine participates in many biological functions, since it is necessary for the synthesis of glutathione, purines, creatine, porphyrins of heme, and primary bile salts. Theoretical calculations have estimated that glutathione synthesis represents approximately 38% of the glycine flux necessary for the synthesis of the metabolites mentioned above; this flow amounts to approximately 1.5 g of glycine/day [4]. In comparison, Meléndez-Hevia et al. [4] estimated that 12 g/day of glycine is needed for collagen synthesis and 1 g/day for the synthesis of other proteins in the body.

It was suggested that glycine availability may be the limiting factor for glutathione synthesis (Figure 3) [130]. Indeed, the tissue glycine concentration is lower than the Michael constant (Km) of glutathione synthase. Thus, in some situations, the availability of glycine may be too low to maintain an adequate synthesis rate of glutathione, especially in metabolic diseases characterized by enhanced oxidative stress [131]. This is associated with an increase in tissue levels of γ-glutamylcysteine, which is converted by the γ-glutamyl cyclotransferase to cysteine and 5-oxoproline, the latter metabolite being excreted in the urine.

Several metabolic studies have suggested that glycine may play a role in insulin resistance and T2DM through alteration in glutathione synthesis rate. Glutathione concentration was 60% lower in erythrocytes of patients with T2DM compared to controls [132]. This was notably associated with a 50% lower absolute synthesis rate of glutathione [132,133]. Limited availability of glycine may play a key role in that alteration, since dietary supplementation with glycine is able to increase the glutathione levels in tissues of several animal models and in humans [124,134]. For instance, glycine supplementation (0.1 g/kg/day) for 14 days in association with N-acetylcysteine overcame the deficiency in glutathione synthesis in patients with T2DM [134] and in elderly HIV-infected patients [124]. In an in vitro setting, improving the transport of glycine in diabetic β-cells has been shown to increase glutathione synthesis and protect against oxidative stress [135]. Finally, a modeling approach in 86 patients with varying degrees of hepatic steatosis pointed out a causative involvement of reduced de novo glutathione synthesis in NAFLD, which was in part overcome after supplementation of glutathione precursors [32].

### 6.2. The Role of Glycine in Heme Biosynthesis

In mammalians, heme production requires three substrates and a multistep biosynthetic pathway involving eight enzymes. The first and rate-controlling reaction of heme biosynthesis is the mitochondrial formation, catalyzed by ALA synthase, of delta-aminolevulinic acid (ALA) from succinyl CoA, generated within the tricarboxylic acid cycle, and glycine [136].

The biosynthesis of one heme molecule requires eight molecules of glycine and one iron atom, and an inadequate supply of glycine to erythroid cells, the lineage in which heme biosynthesis is predominant, leads to a decreased production of heme. Genetic disruption of GLYT1, a glycine transporter 1 in mice, yielded very early postnatal death (1-day old) due to the development of a microcytic anemia [137]. The finding that Glyt1-deficiency causes a lethal phenotype further supports the notion that glycine uptake is necessary, and glycine endogenous production is insufficient, in cell types requiring high amounts of the amino acid [137].

Importantly, heme biosynthesis is crucial for ensuring mitochondrial protein stability and function, such as complex I and III of the electron transfer chain [138]. Decreased heme content could participate in mitochondrial dysfunction and reduced respiratory activity [138].

### 6.3. Importance of Glycine Conjugation and Urinary Excretion

As discussed above, glycine is important for detoxification of certain intermediaries accumulated in excess. This is particularly crucial in obesity and associated metabolic disorders, as synergistic alterations in fatty acid and BCAA metabolism have been reported to raise production of acyl groups [78,139]. Urinary metabolomics profiling revealed lower urinary excretion of acylglycines in patients with obesity compared to lean individuals [140], as well as in diet-induced obese mice compared to lean animals [141]. In addition, improving insulin sensitivity via physical exercise in patients with obesity [140] or by administration of a BCAA restricted diet in diet-induced obese mice [114] increases several glycine conjugates in the urine. This suggests that the glycine conjugation processes may be less efficient in obesity and associated metabolic disorders. Yet, the impact of glycine supplementation on these processes still remains to be investigated.

### 6.4. The Key Role of Glycine in the One-Carbon Metabolism

Glycine metabolism is a key regulator of methyl donors. In the liver, it participates in the regulation of the ratio between S-adenosylmethionine (SAM) and S-adenosylhomocysteine (SAH), as it is the substrate for the enzyme glycine-N-methyltransferase (GNMT) [26,142]. The main biological role of GNMT is to regulate the level of SAM, as it uses SAM and glycine to produce sarcosine. Since SAM is involved in methyl group transfer necessary for many biological processes and maintenance of the genetic stability [5,6], the regulation of GNMT activity is essential for metabolic homeostasis. In addition, GNMT promotes homocysteine re-methylation by the folate-dependent enzyme, methionine synthase, by increasing hepatic content in 5-methyl-THF [143].

GNMT activity is inhibited by 5-methyl-THF produced from CH2-THF reductase [144]. Thus, folate deficiency induces GNMT activity, whereas choline and methionine deficiency inhibits it (Figure 3). GNMT activity is also regulated at the transcriptional level [145]. It is worth noting that over 4 decades ago, GNMT expression was shown to be increased in ageing animals [142]. By contrast, decreased GNMT expression is observed in NAFLDs, both in animal models receiving a high-fat diet [48] and in patients with hepatic steatosis [49]. Furthermore, DNA hypermethylation represses GNMT gene expression in the pathogenesis of NAFLD [146] and hepatocellular carcinoma [145,147]. The loss of GNMT in hepatocellular carcinoma may favor the appearance of aberrant DNA methylation patterns on some promoters [145]. Hence, in Gnmt^−/−^ mice [148,149], hepatic concentrations of methionine and SAM and the ratio of SAM/SAH were multiplied by 7, 36, and 100, respectively, as compared to wild-type animals, suggesting a significant increase in various cellular methylation reactions. Older Gnmt^−/−^ mice develop hepatocellular carcinoma associated with hypermethylation of RASSF1 (Ras association domain family member 1) and SOCS2 (suppressor of cytokine signaling 2) promoters, and silencing of the respective genes. Those two proteins act to inhibit the Ras and JAK/STAT signaling pathways, which are involved in cell proliferation and tumor formation [6].

Gnmt^−/−^ mice, particularly females, also exhibit glucose intolerance and insulin resistance compared to wild-type animals [150]. Very interestingly, a recent study showed that the absence of GNMT causes metabolic reprogramming, whereby nutrients are channeled from glucose formation to pathways that demand high SAM concentration, namely polyamine synthesis and catabolism, trans-sulfuration, and de novo lipogenesis. As a consequence, GNMT mice^−/−^ are characterized by reduced gluconeogenesis due to a decrease in precursors, and hepatic steatosis due to enhanced citrate availability [151].

### 6.5. Glycine as a Neurotransmitter

Glycine is involved in regulating the systemic level of amino acids used as neurotransmitters, whose accumulation in the central nervous system could be neurotoxic [62]. It is a glycine receptor agonist (GlyRs) [152] and an antagonist of the NMDA receptor [153] in the central nervous system. Structure and function of these receptors are reviewed in previous studies [12,63]. In vivo activation of the NMDA receptor by glycine in the dorsal vagal complex was shown to decrease hepatic glucose production, while inhibition of the receptor and hepatic vagotomy neutralized this effect [154]. Furthermore, activation of the NMDA receptor has been involved in controlling food intake, its inhibition leading to an increase in food intake [155]. In another respect, mice fed a high-fat diet were shown to exhibit an increased expression of GlyRs in the hypothalamus, suggesting that the receptor is involved in the central regulation of orexigenic signals in obesity [156].

GlyRs are also expressed in peripheral, non-neuronal tissues and immune cells [12,63]. First, glycine is a well-known secretagogue of key hormones in glucose homeostasis, i.e., GLP-1 [157], insulin, and glucagon [122,123,126]. Explorations on islets of human donors showed that β-cells express the receptor GlyR, especially the GlyRα1 subunit, and the glycine transporters GlyT1 and GlyT2 [158]. GlyR activation by glycine promotes membrane depolarization and insulin secretion in islets of donors without T2DM. The decrease in GlyR expression in β-cells of donors with T2DM has been associated with a disruption of glycine-induced insulin secretion [158]. Second, in macrophages, lymphocyte T, and neutrophils, GlyR activation suppresses the production of pro-inflammatory cytokines by hyperpolarizing the immune cells, thus supporting anti-inflammatory properties [159].

## 7. Conclusions

Glycine is now recognized as a relevant plasma marker for metabolic diseases associated with obesity. It is, thus, considered a promising amino acid for improving metabolic health. However, it is still unclear whether the decline in glycine levels is causatively involved in the pathogenesis of metabolic disorders, particularly glucose intolerance, insulin resistance, and T2DM. Overall, the literature suggests that the key metabolic pathways of glycine can be differentially altered according to metabolic disorders (Figure 4). For example, GNMT deficiency is well described in NAFLD but not in T2DM. Understanding the differences and similarities between T2DM and NAFLD in relation to glycine metabolism will be of the greatest interest to improve our understanding of the relationship between the two pathologies. Likewise, it will be extremely relevant to decipher the role of glycine metabolism in cancers, in which glycine metabolism is also fundamental, not least to govern the methylation status of cancer cells [160,161].

If the benefit of glycine for improving metabolic health is confirmed, the mechanisms underlying the long-term effects of glycine supplementation will need to be fully understood. Doses and modes of administration of the amino acid will have to be determined in phase I dose escalation studies to ensure the efficacy and safety of dietary supplementation and clinical treatment. In that context, as the importance of the gut microbiota in determining the availability of glycine becomes more obvious, promising prospects are expected in the development of pre- and probiotics quantitatively affecting glycine bioavailability to the host.

## Figures and Tables

**Figure 1 nutrients-11-01356-f001:**
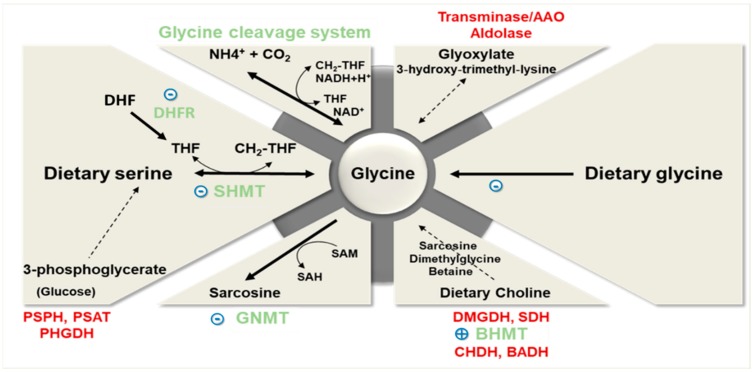
**Main dietary sources and metabolic pathways of glycine**. (Enzymes with a quantitatively prominent significant physiological role are presented in gray). Note: AAO = D-amino acid oxidase; BHMT = betaine-homocysteine S-methyltransferase; CHDH = choline dehydrogenase; DHF = dihydrofolate; DHFR = dihydrofolate reductase; DMGDH = dimethylglycine dehydrogenase; GNMT = glycine N-methyltransferase; PHGDH = phosphoglycerate dehydrogenase; PSAT = phosphoserine aminotransferase; PSPH = phosphoserine phosphatase; SAM = S-adenosylmethionine; SAH = S-adenosylhomocysteine; SDH = sarcosine dehydrogenase; SHMT = serine hydroxymethyltransferase; THF = tetrahydrofolate; CH_2_-THF = N^5^, N^10^-methylene tetrahydrofolate. Labels in blue evidence the obesity-associated alterations in the expression or activity of the main enzymes determining glycine availability (for details see text in Section 2). Dietary glycine availability and uptake by the organism is regulated by the microbiota and gut metabolism (for details, see text in Section 3).

**Figure 2 nutrients-11-01356-f002:**
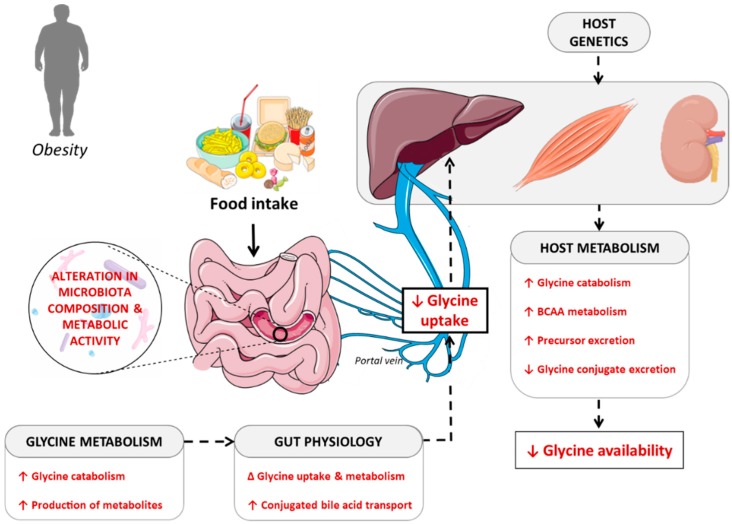
**Potential mechanisms contributing to systemic glycine deficiency during metabolic diseases associated with obesity**. Glycine dietary intake may not be the main determinant of glycine availability for the organism. Interactions between the food matrix and the intestinal microbiota influence the bacterial composition and metabolic capacity of the latter, thus modifying its ability to use glycine and produce metabolites derived from glycine. Alterations in glycine availability or microbial metabolites can modulate the expression of genes in intestinal compartments and impact the ability of the intestinal epithelium to take up glycine. Finally, interactions between the host genetics and physiology and the amount of glycine driven through the portal vein determine the fate of glycine, its bioavailability for the whole body, and its consequences on the host metabolism.

**Figure 3 nutrients-11-01356-f003:**
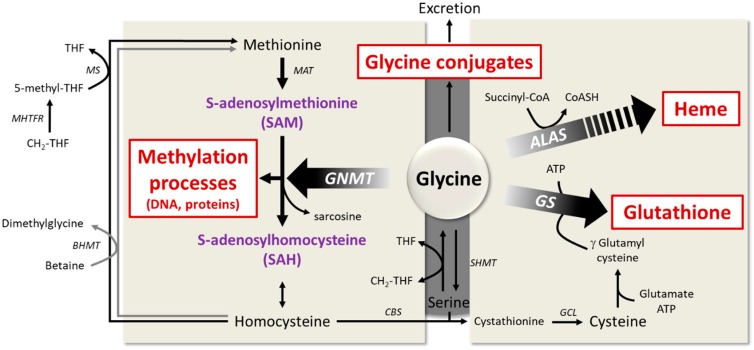
**Main pathways involving glycine in host metabolism**. Note: ALAS = delta-aminolevulinic acid synthase; BHMT = betaine-homocysteine S-methyltransferase; CBS = cystathionine β-synthase; CGL = cystathionine γ-lyase; GNMT = glycine N-methyltransferase; GCL = glutamate–cysteine ligase; GS = glutathione synthase; MAT = methionine adenosyltransferase; MS = methionine synthase; SAM = S-adenosylmethionine; SAH = S-adenosylhomocysteine; SDH = sarcosine dehydrogenase; SHMT = serine hydroxymethyltransferase; THF = tetrahydrofolate; CH_2_-THF = N^5^, N^10^-methylene tetrahydrofolate; 5-methyl-THF = 5-methyltetrahydrofolate.

**Figure 4 nutrients-11-01356-f004:**
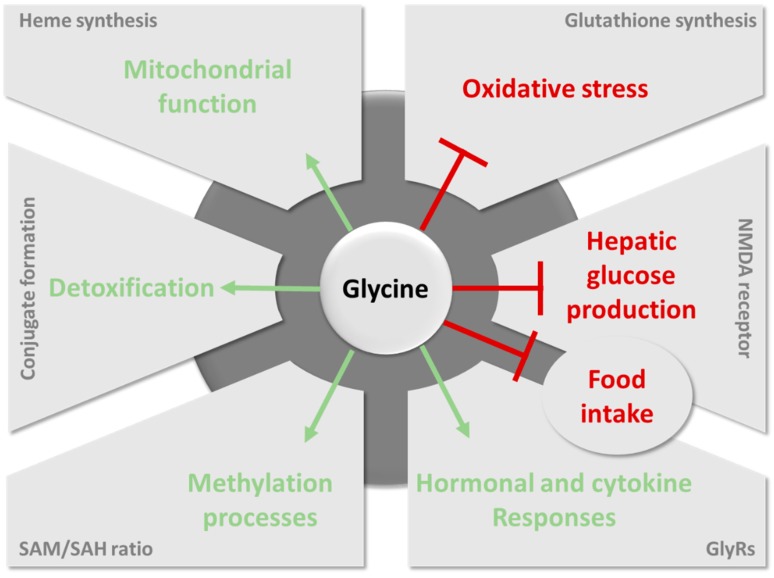
**Main pathways involving glycine in health benefits**. Metabolic benefits mediated by glycine include the inhibition of oxidative stress via increased glutathione biosynthesis, an inhibitory effect on gluconeogenesis and food intake via activation of the NMDA receptor, curbing the overload. Glycine also exerts positive effects on mitochondrial activity via heme biosynthesis, detoxification processes via urinary excretion of glycine conjugates, and regulation of hormonal (enhanced secretion of key hormones in glucose homeostasis) and cytokine (reduced production of pro-inflammatory cytokines) responses via activation of GlyRs. Finally, glycine impinges the SAM biosynthetic process, decreasing the availability of methyl-donors, and thus regulating methylation. Favorable pathways induced by glycine are green; the harmful pathways inhibited by glycine are red. Note: NMDA = N-methyl-D-aspartate; GlyRs = glycine receptors; SAM = S-adenosylmethionine; SAH = S-adenosylhomocysteine.

**Table 1 nutrients-11-01356-t001:** Summary of key observational studies correlating glycine serum or plasma levels to metabolic disease.

Study Group	Health Status	Glycine Concentration (µmol/L)		Level of Significance	Reference
		Control Group	Study Group		
20 control subjects, 15 subjects with obesity with NAFLD	Controls vs. obesity with NAFLD	Mean: 205.9 ± 9.7	Mean obesity with NAFLD: 179.2 ± 7.6	*p* = 0.03	[9]
The observational, prospective cohort PPSDiab: 151 women with gestational diabetes or normoglycemia during pregnancy	NGT vs. PGT	Median NGT: 272.6	Median PGT: 224.6	*p* < 0.01	[14]
399 nondiabetic adults	IS vs. IR	*NA*	0.85 fold vs. controls	*p* = 2.79 × 10^−11^	[15]
124 adults (63 European American and 60 African American)	IS vs. IR and T2DM	Mean IS: 306.8	Mean IR: 257.0	*p* < 0.01 for the two comparisons vs. IS	[16]
			Mean T2DM: 246.8		
			[Glycine concentration was correlated to GDR in a hyperinsulinemic-euglycemic clamp]		
64 adults	sex-matched groups for BMI [lean vs. morbid obesity] and risk of developing T2DM [IS vs. IR]		Glycine concentration is negatively associated to fasting insulin and HOMA-IR	R = −0.51, *p* = 0.0017; and R = −0.49, *p* = 0.0033	[17]
Framingham Heart Study (*n* = 1015) and the Malmö Diet and Cancer Study (*n* = 746)	45% of individuals meeting the criteria for metabolic syndrome	Mean NGT = 270	Mean PGT = 220	*p* = 0.0005	[18]
73 control subjects, 10 subjects with obesity	Controls vs. obesity	Mean: 223.7 ± 33.0	Mean: 197.9 ± 41.4	*p* = 0.027	[19]
51 healthy control subjects; 31 overweight or obese subjects; 52 subjects with T2DM	Controls vs. obesity and T2DM Men and women were analyzed separately	Mean men: 211 ± 30	Mean men with obesity: 186 ± 30	*p* < 0.05 for all comparisons vs. controls	[20]
			Mean men with T2DM: 187 ± 44		
		Mean women: 231 ± 67	Mean women with obesity: 203 ± 48		
			Mean women with T2DM: 184 ± 48		

Note: NGT = Normal glucose tolerance; PGT = pathological glucose tolerance; IS = insulin sensitive; IR = insulin resistant; NA = not available; T2DM = type 2 diabetes; GDR = glucose disposal rate; BMI = body mass index; HOMA-IR = Homeostatic Model Assessment; NAFLD = non-alcoholic fatty liver diseases, PPSDiab: Prediction, Prevention and Sub-classification of type 2 Diabetes.

**Table 2 nutrients-11-01356-t002:** Summary of preclinical and clinical studies that have evaluated the health impact of glycine or betaine dietary supplementation.

Population	Health Status	Dose and Duration	Health Impacts of Glycine Supplementation	Reference
**Glycine dietary supplementation**				
Clinical studies				
Adult humans:	Healthy patients	Single oral morning dose of 5 g glycine +/− 25 g glucose vs. water +/− 25 g glucose	Improves insulin response and glucose tolerance in response to glucose ingestion	[122]
4 Women				
5 Men				
Age: 21 to 52 y				
Adult humans:	Healthy lean patients with first degree relatives of T2DM	Single oral morning dose of 5 g glycine vs. magnesium oxide (placebo)	Improves insulin response, measured during an euglycemic-hyperinsulinemic clamp; No significant alteration in insulin action	[123]
8 Women				
4 Men				
Age: 23.7 ± 4.1 y				
Adult humans:	Patients with MetS (NCEP/ATP III criteria)	15 g glycine/day (3 times 5 g/d) dissolved in water vs. starch (placebo) for 3 months	Improves systolic blood pressure in men; Protects against oxidative damages determined from antioxidant enzymes activity in erythrocytes and leukocytes, and thiobarbituric acid reactive substances (TBARS) in plasma	[125]
29 Women				
23 Men				
Age: 35 to 65 y				
Elderly patients:	Patients with HIV	1.33 mmol glycine/kg/day with 0.81 mmol/kg/day N-acetylcysteine for 14 days	Improves insulin sensitivity, measured by hyperinsulinemic-euglycemic clamp before and after supplementation	[124]
9 Men				
Age: 56.1 ± 1.0 y				
*Preclinical studies*				
Male Sprague Dawley rats: *n* = 48	High fat/high sucrose feeding vs. standard chow for 24 weeks	3.5 g glycine/kg/day in water vs. water (placebo) for 24 weeks	Improves hepatic steatosis assessed histologically	[128]
Age: *NA*				
Male KK-Ay mice: *n* = 5/group	Animal model of obesity and T2DM	Semisynthetic diet containing 5% glycine vs. casein (placebo) for 4 weeks	Improves hepatic steatosis assessed histologically Improves glucose tolerance measured during a glucose tolerance test	[129]
Age: 7 weeks				
***Betaine dietary supplementation***				
*Clinical studies*				
Adult humans:	patients with obesity and pre-diabetes	3.30 g betaine, twice daily for 10 days, followed by 4.95 g twice daily for 12 weeks vs. microcrystalline cellulose (placebo)	No major effects on glucose homeostasis (euglycemic hyperinsulinemic clamp) and liver fat deposition	[127]
8 Women				
20 Men				
Age: 21 to 70 y				
*Preclinical studies*				
Female	High-fat feeding for 13 weeks	1% weight/volume betaine, in water vs. water for 1 week	Improves insulin resistance and glucose homeostasis measured using glucose/insulin tolerance tests	[126]
Kunming				
Mice: *n* = 40				
Age: 6 weeks				
Male C57Bl6 mice: *n* = 24	High-fat feeding for 16 weeks	1% weight/volume betaine, in water vs. water for 1 week	Improves insulin resistance and glucose homeostasis measured using glucose/insulin tolerance test and euglycemic hyperinsulinemic clamp; Reduces liver fat deposition quantified on chloroform-methanol extracts	[87]
Age: *NA*				
Male C57BL6/N mice: *n* = 46	High-fat feeding for 12 weeks, methyl-donor supplementation was given during the last 4 weeks	15 g/kg betaine, 15 g/kg choline chloride, 7.5 g/kg methionine, 15 mg/kg folic acid, 1.5 mg/kg vitamin B12, 150 mg/kg ZnSO4	Prevented the progression of hepatic steatosis Increases phosphorylation of AMPK-α together with enhanced β-HAD activity, suggesting increased fatty acid oxidation	[34]
Age: 8 weeks				

Note: MetS = Metabolic syndrome; T2DM = type 2 diabetes; NA = not available.

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
