# Peer review of "Glycine Metabolism and Its Alterations in Obesity and Metabolic Diseases"

_nutrients, 2019, doi:10.3390/nu11061356_

Round 1
Reviewer 1 Report
In this review, authors presented glycine metabolism and its alterations in metabolic disorders. On the whole, this review showed the recent advances in this research area, and had important scientific significance, to some extent. This manuscript was well organized in terms of logicality, and also gave some detailed statement in every section. In my opinion, it was a very nice review. I only have some minor comments, as follows:
1. Figure 1 showed the main dietary sources and metabolic pathways of glycine. Can you add some information or draw a new figure to describe the changes in glycine endogenous biosynthetic and catabolic pathways in subjects with metabolic disorders?
2. Can you draw a figure to summarize how glycine supplement improve metabolic disorders, especially obesity and insulin resistance, if it is possible?
Could you draw a figure and add some references to describe how Glycine affects metabolic disorders (especially obesity and insulin resistance)-related signaling pathways?
Author Response
Answers to Reviewer 1
In this review, authors presented glycine metabolism and its alterations in metabolic disorders. On the whole, this review showed the recent advances in this research area, and had important scientific significance, to some extent. This manuscript was well organized in terms of logicality, and also gave some detailed statement in every section. In my opinion, it was a very nice review. I only have some minor comments, as follows:
We thank the reviewer for the positive evaluation of our manuscript. As suggested, also by second
independent referee, we have now improved the figures.
1. Figure 1 showed the main dietary sources and metabolic pathways of glycine. Can you add some
information or draw a new figure to describe the changes in glycine endogenous biosynthetic and
catabolic pathways in subjects with metabolic disorders?
Figure 1 has been completed and alterations in the expression and/or activity of the main enzymes
determining glycine availability in obesity have been highlighted.
2. Can you draw a figure to summarize how glycine supplement improve metabolic disorders,
especially obesity and insulin resistance, if it is possible?
Could you draw a figure and add some references to describe how Glycine affects metabolic
disorders (especially obesity and insulin resistance)-related signaling pathways?
As suggested by the reviewer, we have now added a new figure (figure 4) summarizing the impact of glycine on different biological events, and the beneficial effect brought about by the amino acid.

Reviewer 2 Report
Main Points
Overall this is a quite well written and comprehensive review for the areas of human glycine metabolism covered. However, there are several general points I would like the authors to address before I would regard this review as suitable for publication:
1. You absolutely must replace the majority of citations of other reviews with original research citations. This is particularly true of the introduction. I will indicate some examples in the relevant sections below, but please go through the manuscript careful and do this. There are only two instances where a review should cite an already existing review: when a very general and long-established point is being made (a good example is citation 21); or when you are providing information that is necessary but peripheral to the review’s main topic. This over-reliance on other reviews in certain sections is reflected, I think, in the low number of total citations (i.e. 124). I would have expect far more than 124 references for a review on such a large topic that was doing due justice to all the primary research.
2. Another general point on citations. There also seems to be a lack of comprehensive primary research article citation in several sections, particularly section 2. Please take the time to find the primary citation(s) to provide evidence the information being provided. If the point was elucidated over several papers cite them all. In addition, I note many of the describe biochemical reactions detailing the anabolism and catabolism of glycine are not cited at all. Indeed, there are entire stretches of text (e.g. section 2.2 last two paragraphs) where no citations are given at all. Please correct with original research citations.
3. In the abstract or introduction you give no rationalization for why another review on glycine metabolism is required. There have been numerous review in the past 5 years on glycine metabolism and, specifically, its role in metabolic disorders e.g. PMID 28337245, 23615880, 27292783, 29094215). What is your review adding to the field in terms of summary/synthesis that these other reviews have not? I can see you have emphasized the role of glycine in metabolic diseases in your review- fine but outline what new research points have not yet by covered in existing reviews.
4. To follow up on the previous point and, perhaps, help you to differentiate this review, there are several topics within glycine metabolism that you have not covered that have advanced in the last few years and some of which are not covered, or covered in a limited fashion, in other reviews. This includes:
· The role of glycine in Heme biosynthesis and the fact it is the rate limiting step of this process (see PMID 28495919, 27476175 for good recent publications and review PMID 23720291 on heme synthesis for starting points).
· Major uptake proteins (transporters for glycine) involved in intestinal dietary intake and re-absorption from kidney as well as major glycine transporters in human organs (i.e. SLC36 gene family (PATs) and SLC6 A20 transporter in intestine and kidney, SLC38 family (transporters (SNATs) in other organs) need to be added to the dietary intake section and (for re-absorption of glycine) the ‘glycine conjugation and excretion’ section Nutrient transporters are the first stage of metabolism by controlling entry into cells.
· Glycine’s role as a neurotransmitter and in pain sensation via glycine receptors and transporters. You have a small sub-section on thesis in section 6 but this presents only the bare minimal of information and cites only very broad reviews. If you are going to include a sub-section on this topic provide the reader with something useful or leave it out altogether and mention in passing as ‘other roles of glycine not covered in this review’.
These sections could be incorporated into section 6 as sub-sections and should be treated with at least a paragraph each.
5. Section 6 is really part of section 2.2 and 2.3. I suggest removing it altogether and incorporating all these sub-sections within section 2. After all, together these sub-sections describe the various metabolic fates and utilizations of glycine and should be grouped.
Abstract & Introduction
1. Following on from the main point above, there are numerous sentences in the introduction where original research articles should be cited but which cite generic reviews E.g. The sentences cited with references 1 and 2, and the sentence ending ‘…is not life-threatening.’ The last No citation at all is given for the last example – where is the evidence for this claim? As I wrote above there are numerous examples like this throughout the manuscript but particularly in the first two sections.
2. The use of ‘simplest’, ‘simplicity’ to describe glycine. These are not clearly defined chemical terms that I know of. What do you mean by glycine being ‘simple’ – in terms of atomic composition, lability? Please avoid subjective terminology and remove these terms from the abstract and introduction. No chemical is inherently ‘simple’ unless a specific property is defined. Glycine is definitely not simple in terms of its metabolic utilization and amphipathic properties.
Section 2
1. Reference 22 seems not to be the original research illustrating that other metabolites contribute much less than serine to de novo glycine synthesis. Please check and change if required.
2. In section 2.1, the paragraph beginning ‘In healthy individuals, glycine is inter-convertible…’ This implies that inter-conversion between glycine and serine is hampered or does not occur in ‘unhealthy’ individuals. If so please elaborate with references or edit to remove confusion.
3. Original research citations required for large sections on enzymes: SHMT1, AGAT, and 4 mitochondrial de novo glycine synthesis enzymes. Many citations to add!
4. Section 2.2.: the sentence beginning ‘Using 13C-glycine infusion, it was observed…’ Several points: firstly, replace ‘metabolic disorders’ with obesity – the study was conducted on obese rats, correct? Then the claim cannot be made that glycine to serine conversion involves general metabolic disorders. Secondly, this sentence really belongs in section 3.
5. Reference 28 is a review, replace with original research citations. There is a vast literature on the early research of dietary amino acids, including glycine, that is well summarized in Matthew’s 1991 book ‘Protein absorption’ – you can utilize that book as a starting point to find original research I would suggest.
Section 3 and 4
1. I haven’t too much to add concerning these sections as they are your field, your hypothesis and it is well written. However, I would strongly suggest you modify and greatly improve Figure 2. You have simply duplicated the same panel and changed the text in the circle – this is neither very informative nor does it really outline what you have said in the text. A good review figure should do both of these things. I would consider putting in a lot more effort to create a figure which summarizes the knowledge on the link between dietary intake/obesity, gut microbiota and glycine metabolism and availability as you have in the text. Also make the figure bigger – it should be a centerpiece of the review.
Section 5
2. Rather that continually refer to ‘Table 2’ you can simply mention at the beginning of the relevant sections that Tables 1 and 2 summaries the major studies outline in the following sections.
3. I do not think you have given the full name for DMGDH (I presume this is Dimethylglycine Dehydrogenase?) anywhere. Please make sure all enzymes and protein acronyms, when introduced for the first time, are given their full names – they can be abbreviated thereafter. Check for all abbreviations. In addition, where is the abbreviation list – there should be one.
4. The clause ‘…glycine supplementation may also show benefits in NAFLDs but no clinical data…’ I have no idea what this refers to or how you have made this link. In fact you state there is no clinical data to support the notion, is there other data? If you have presented a link between NAFLD and glycine in another section than this statement is clearly out of place. Alternatively, if you are trying to make a new point you need some information and context.
5. The final paragraph is also out of place. You have already introduced glucagon and glycine at the start of sub-section 4.3. Move this to that section if you need to provide more summary. In addition the information you do provide is very generic and provides no real insight to the effect of glycine on glucose homeostasis. If you cannot review it comprehensively and synthesis the research than leave it out altogether.
Author Response
Answers to Reviewer 2
Main Points
Overall this is a quite well written and comprehensive review for the areas of human glycine
metabolism covered. However, there are several general points I would like the authors to address
before I would regard this review as suitable for publication.
We wish to thank the reviewer for her/his positive overall appraisal of our manuscript. We also
acknowledge the reviewer constructive criticism, which we have tried to address as thoroughly as
possible, as it is presented below in the point-to-point discussions. We have made an effort to be as
complete as possible in the integration on the reviewer’s comments.
1. You absolutely must replace the majority of citations of other reviews with original research
citations. This is particularly true of the introduction. I will indicate some examples in the relevant
sections below, but please go through the manuscript careful and do this. There are only two
instances where a review should cite an already existing review: when a very general and longestablished point is being made (a good example is citation 21); or when you are providing
information that is necessary but peripheral to the review’s main topic. This over-reliance on other
reviews in certain sections is reflected, I think, in the low number of total citations (i.e. 124). I
would have expect far more than 124 references for a review on such a large topic that was doing
due justice to all the primary research.
We agree on the reviewer’s comment. The overall literature on glycine metabolism and its
disturbances in metabolic disease is, of course, far larger that wat we covered with our 124 references.
In this revised version, we diminished the overreliance on citing previous reviews to favor citations of primary research and we have added 47 additional references. While we do not wish to deter any
merit from previously published reviews, we would like to point out that our manuscript provides a
significant update on the literature, with a substantial amount of references to literature published
from 2015
2. Another general point on citations. There also seems to be a lack of comprehensive primary
research article citation in several sections, particularly section 2. Please take the time to find the
primary citation(s) to provide evidence the information being provided. If the point was elucidated
over several papers cite them all. In addition, I note many of the describe biochemical reactions
detailing the anabolism and catabolism of glycine are not cited at all. Indeed, there are entire stretches of text (e.g. section 2.2 last two paragraphs) where no citations are given at all. Please correct with original research citations.
As mentioned in our previous answer, we have made effort to cite primary references and to add
references when missing.
3. In the abstract or introduction you give no rationalization for why another review on glycine
metabolism is required. There have been numerous review in the past 5 years on glycine metabolism and, specifically, its role in metabolic disorders e.g. PMID 28337245, 23615880, 27292783, 29094215).
What is your review adding to the field in terms of summary/synthesis that these other reviews have
not? I can see you have emphasized the role of glycine in metabolic diseases in your review- fine but outline what new research points have not yet by covered in existing reviews.
We try to be clearer on the scope and novelty of our review, yet without overemphasizing as of course other reviews appeared and more will be published in the future, as the body of primary research increases.
4. To follow up on the previous point and, perhaps, help you to differentiate this review, there are
several topics within glycine metabolism that you have not covered that have advanced in the last few years and some of which are not covered, or covered in a limited fashion, in other reviews. This
includes:
· The role of glycine in Heme biosynthesis and the fact it is the rate limiting step of this process
(see PMID 28495919, 27476175 for good recent publications and review PMID 23720291 on heme
synthesis for starting points).
· Major uptake proteins (transporters for glycine) involved in intestinal dietary intake and reabsorption from kidney as well as major glycine transporters in human organs (i.e. SLC36 gene family (PATs) and SLC6 A20 transporter in intestine and kidney, SLC38 family (transporters (SNATs) in other organs) need to be added to the dietary intake section and (for re-absorption of glycine) the ‘glycine conjugation and excretion’ section Nutrient transporters are the first stage of metabolism by
controlling entry into cells. ·
Glycine’s role as a neurotransmitter and in pain sensation via glycine receptors and transporters.
You have a small sub-section on thesis in section 6 but this presents only the bare minimal of
information and cites only very broad reviews. If you are going to include a sub-section on this topic
provide the reader with something useful or leave it out altogether and mention in passing as ‘other
roles of glycine not covered in this review’.
These sections could be incorporated into section 6 as sub-sections and should be treated with at least a paragraph each.
We acknowledge the recent advances in these biological phenomena that integrate glycine
metabolism. Paragraphs have been added in the revised version in section 2 (glycine transporters) and 6 (heme biosynthesis, glycine as a neurotransmitter). Figure 3 has been corrected accordingly.
5. Section 6 is really part of section 2.2 and 2.3. I suggest removing it altogether and incorporating
all these sub-sections within section 2. After all, together these sub-sections describe the various
metabolic fates and utilizations of glycine and should be grouped.
We thank the reviewer for this comment. Section 2 “ Glycine dietary intake and metabolism” provides an overview of glycine anabolic and catabolic pathways, with subsections 2.2 and 2.3 discussing glycine catabolism and excretion respectively. As it stands, section 2 aims to give an overview of glycine metabolic routes without the intention to delve into the pathogenesis of metabolic disorders. We believe merging section 6 within section 2 would render section 2 too ample and in some instances too much explicatory of phenomena discussed elsewhere in the review, for example section 3, in which the connection between altered glycine levels and metabolic disease starts to be discussed.
Abstract & Introduction
1. Following on from the main point above, there are numerous sentences in the introduction where
original research articles should be cited but which cite generic reviews E.g. The sentences cited
with references 1 and 2, and the sentence ending ‘…is not life-threatening.’ The last No citation at
all is given for the last example – where is the evidence for this claim? As I wrote above there are
numerous examples like this throughout the manuscript but particularly in the first two sections.
Use of references to previous reviews has been motivated, in part, by the editorial guidelines and
requirements for review style. We agree, however, that landmark original reports should be reported.
Now, we have either supplemented the citation to a review with the most important original paper or,
altogether, removed the reference to a review to directly cite research.
2. The use of ‘simplest’, ‘simplicity’ to describe glycine. These are not clearly defined chemical terms that I know of. What do you mean by glycine being ‘simple’ – in terms of atomic composition, lability?
Please avoid subjective terminology and remove these terms from the abstract and introduction. No
chemical is inherently ‘simple’ unless a specific property is defined. Glycine is definitely not simple interms of its metabolic utilization and amphipathic properties.
We thank the reviewer for this comment. We now abide to a more formal discussion of glycine’s
chemistry as follows: Abstract: "Glycine is the smallest proteinogenic amino-acid, harboring a
hydrogen atom as side-chain.” We also remove the period “In spite of its chemical simplicity”.
The introduction opening paragraph has been modified as follows: “Glycine is the protein-forming
amino acid with the smallest molecular weight (NH2‐CH2‐COOH, 75.067 g/mol).” The lack of chemical features in the glycine’s side chain allows it to integrate both hydrophilic and hydrophobic
environments within the polypeptide chain.”
Section 2
1. Reference 22 seems not to be the original research illustrating that other metabolites contribute
much less than serine to de novo glycine synthesis. Please check and change if required.
Reference 22 (now #4) is indeed an original work.
2. In section 2.1, the paragraph beginning ‘In healthy individuals, glycine is inter-convertible…’ This
implies that inter-conversion between glycine and serine is hampered or does not occur in ‘unhealthy’individuals. If so please elaborate with references or edit to remove confusion
We accept the reviewer comment. As section 2 presents the physiological and biochemical aspects ofglycine metabolism, we removed the reference to the healthy state.
3. Original research citations required for large sections on enzymes: SHMT1, AGAT, and 4
mitochondrial de novo glycine synthesis enzymes. Many citations to add!
We thank the reviewer for the constructive comment. We have improved the section and added
original studies.
4. Section 2.2.: the sentence beginning ‘Using 13C-glycine infusion, it was observed…’ Several points:firstly, replace ‘metabolic disorders’ with obesity – the study was conducted on obese rats, correct? Then the claim cannot be made that glycine to serine conversion involves general metabolic disorders. Secondly, this sentence really belongs in section 3.
As the study was done on obese rats, we have replaced “metabolic disorders” with “obesity” as
correctly suggested by the reviewer. As for moving the discussion of this study to section 3, section 2 deals with the glycine metabolism under physiological condition. In a way, the discussion on 13Glycineinfused obese rats is in part deviating from the main topic of the section but, somehow, it alludes to the presentation of altered glycine metabolism in the subsequent paragraph. We would like, therefore, to maintain the reference in its original position.
5. Reference 28 is a review, replace with original research citations. There is a vast literature on the
early research of dietary amino acids, including glycine, that is well summarized in Matthew’s 1991
book ‘Protein absorption’ – you can utilize that book as a starting point to find original research I would suggest.
Reference 28 (now #35) is indeed a clinical study.
Section 3 and 4
1. I haven’t too much to add concerning these sections as they are your field, your hypothesis and it
is well written. However, I would strongly suggest you modify and greatly improve Figure 2. You
have simply duplicated the same panel and changed the text in the circle – this is neither very
informative nor does it really outline what you have said in the text. A good review figure should
do both of these things. I would consider putting in a lot more effort to create a figure which
summarizes the knowledge on the link between dietary intake/obesity, gut microbiota and glycine
metabolism and availability as you have in the text. Also make the figure bigger – it should be a
centerpiece of the review.
We thank the reviewer for the positive comment. As also requested independently by reviewer 1,
some improvement of the figures was suggested, and it has been attended to.
Section 5
2. Rather that continually refer to ‘Table 2’ you can simply mention at the beginning of the relevant
sections that Tables 1 and 2 summaries the major studies outline in the following sections.
We thank the reviewer for pointing this out. We have modified the manuscript to avoid multiple
referencing to the information contained into the tables.
3. I do not think you have given the full name for DMGDH (I presume this is Dimethylglycine
Dehydrogenase?) anywhere. Please make sure all enzymes and protein acronyms, when
introduced for the first time, are given their full names – they can be abbreviated thereafter. Check
for all abbreviations. In addition, where is the abbreviation list – there should be one.
DMGDH was given in full name in the main text. Furthermore, we acknowledge the importance of
adding a list. We have now added an abbreviation list. This list contains all abbreviations occurring at least twice in the main text. The abbreviations for bile acids, as occurring only once in the text, have been outright removed, as well as abbreviations of the names of certain clinical trials in which the acronym is not widely known. In a few cases, an abbreviation is used only once in the text (as for example the CODING study: Complex Diseases in the Newfoundland Population: Environment and Genetics). In this case, we spell out the abbreviation only in the text. Finally, it shall be noted that in the figure legends, for clarity, we report the abbreviations appearing in the figure.
4. The clause ‘…glycine supplementation may also show benefits in NAFLDs but no clinical data…’ I have no idea what this refers to or how you have made this link. In fact you state there is no clinical data to support the notion, is there other data? If you have presented a link between NAFLD and glycine in another section than this statement is clearly out of place. Alternatively, if you are trying to make a new point you need some information and context.
We believe part of the interest of review articles resides in the fact that the study of existing literature provides a way to develop new questions, hypotheses and experimental studies. Here, specifically, we propose that the beneficial effect of glycine on NAFLDs observed in animal models should next pass the scrutiny in a clinical setting. Our statement was perhaps too forward looking, and we modified it as follows:” Furthermore, the observation that glycine supplementation improves hepatic steatosis in animal models of obesity [125,126] should provide a motivation for human clinical testing”
5. The final paragraph is also out of place. You have already introduced glucagon and glycine at the
start of sub-section 4.3. Move this to that section if you need to provide more summary. In addition
the information you do provide is very generic and provides no real insight to the effect of glycine on
glucose homeostasis. If you cannot review it comprehensively and synthesis the research than leave it out altogether.
We acknowledge that the place for this information was inappropriate. However, we think it is
important to keep it. We moved it end of section 6 in the novel paragraph dealing with glycine
receptors. The information is still generic and supports the importance of GlyR in regulating insulin
secretion be beta-cells.
